# Probiotics Function in Preventing Atopic Dermatitis in Children

**DOI:** 10.3390/ijms23105409

**Published:** 2022-05-12

**Authors:** Caterina Anania, Giulia Brindisi, Ivana Martinelli, Edoardo Bonucci, Miriam D’Orsi, Sara Ialongo, Anna Nyffenegger, Tonia Raso, Mattia Spatuzzo, Giovanna De Castro, Anna Maria Zicari, Carlo Carraro, Maria Grazia Piccioni, Francesca Olivero

**Affiliations:** 1Department of Mother-Child, Urological Science, Sapienza University of Rome, 00161 Rome, Italy; giulia.brindisi@uniroma1.it (G.B.); ivana.martinelli@uniroma1.it (I.M.); bonucciedoardo@gmail.com (E.B.); dorsimiriam@gmail.com (M.D.); sara.ialongo01@gmail.com (S.I.); annanyffe@gmail.com (A.N.); toniaraso@gmail.com (T.R.); spatuzzomattia@gmail.com (M.S.); giovanna.decastro@uniroma1.it (G.D.C.); annamaria.zicari@uniroma1.it (A.M.Z.); carlo.carraro@uniroma1.it (C.C.); mariagrazia.piccioni@uniroma1.it (M.G.P.); 2Pediatric Clinic, Department of Pediatrics, Fondazione IRCSS Policlinico San Matteo, University of Pavia, 27100 Pavia, Italy; francesca.olivero02@universitadipavia.it

**Keywords:** atopic dermatitis, probiotics, prevention, children

## Abstract

Atopic dermatitis (AD) is a chronic inflammatory skin disorder characterized by relapsing eczematous injuries and severe pruritus. In the last few years, the AD prevalence has been increasing, reaching 20% in children and 10% in adults in high-income countries. Recently, the potential role of probiotics in AD prevention has generated considerable interest. As many clinical studies show, the gut microbiota is able to modulate systemic inflammatory and immune responses influencing the development of sensitization and allergy. Probiotics are used increasingly against AD. However, the molecular mechanisms underlying the probiotics mediated anti-allergic effect remain unclear and there is controversy about their efficacy. In this narrative review, we examine the actual evidence on the effect of probiotic supplementation for AD prevention in the pediatric population, discussing also the potential biological mechanisms of action in this regard.

## 1. Introduction

It is now extensively demonstrated that the intestinal microbiota has a pivotal role on the state of health, contributing to maintain a strong and highly functional immune system and the stability of the intestinal barrier. Current data suggest that gut dysbiosis, especially if it happens early in life, contributes to the development of inflammatory conditions including allergies. The microbial infants’ gut colonization starts before birth and continues into childhood through exposure to environmental factors. It is currently known that the mother transfers microorganisms to the newborn through the placenta, intestine, meconium, and vagina [1]. The microbiota of the placenta and umbilical cord is influenced by maternal nutrition and health status [2]. Infant gut microbiota are affected by diverse perinatal conditions including mode of delivery, nutrition, and antibiotic usage. Newborns from vaginal delivery have a greater variety of bacterial flora and a higher amount of *Bifidobacterium*, *Bacteroides,* and *Lactobacillus* compared to those born by caesarean delivery [3,4,5]. Moreover, considering breastfed infants, their gut microbiota is less diverse and dominated by *Bifidobacteria* compared to formula-fed ones, showing a higher proportion of *Firmicutes* [6]. In addition, breastfeeding exerts a basic role in establishing the infant gut microbiota as a result of the bioactive compound contents, which includes human milk oligosaccharides (HMOs). It has been seen that these features regulate the proliferation and maturity of gastrointestinal cells and tend to provide nutrients, probiotics, and IgA, which enhances the immune system [7]. In the first year of life, the gut microbiome changes rapidly, and it is enriched by the interaction with external environment. From the second year of life and beyond, the child’s gut microbiome begins to stabilize and increasingly resembles that of an adult reaching an adult-like microbiome profile in later childhood [8]. The risk of developing allergies has been related to intestinal dysbiosis due to an early derangement in the composition and/or function of the intestinal microbiota already existing in the first months of life. Allergic subjects reveal different gut composition to non-allergic infants and a reduced microbial variety, including a decrease of *Lactobacilli* and *Bifidobacteria* [9,10]. Moreover, it has been underlined that the latest rise in prevalence of allergic diseases including AD may be the result of an early intestinal dysbiosis [11]. AD is the most frequent, chronic, recurrent skin disorder with a broad clinical spectrum, often linked with other allergies, such as food allergy (FA), asthma, and allergic rhinitis (AR). Microbial flora alterations, along with epithelial barrier defects and an impaired immune response, are engaged in the pathogenesis of this disease. Evidence suggests that AD derives from a T-cell imbalance with the predominance of T helper cell type 2 (Th2) differentiation of naïve CD4+ T cells resulting in a greater production of interleukins IL-4, IL-5, and IL-13, which could be locally affected both by the activation of IgE and in eosinophils. Recently, it has been reported that probiotics could be a potential preventive strategy for allergies including AD through the enhancement of epithelial barrier integrity as well as modulating the immune system through the rebalancing of the Th1 and Th2 response on epithelial barrier integrity and the modulation of the immune by rebalancing the Th1 and Th2 response [12]. In this narrative review, we provided an updated overview of the recent evidence on the use of probiotics in AD prevention among children, analyzing the results of recently performed clinical interventions and discussing briefly the possible biological mechanisms linking the probiotic action and AD prevention.

### Methods

The present review provides a look of the recent literature on the efficacy of probiotics for AD prevention. Evidence published in the last ten years has been searched using the PUBMED AND SCOPUS Library. We included as a research strategy the following keywords: Atopic dermatitis prevention AND probiotics children. We also inserted these filters: Full text, 10 years, Humans, English, Child: birth-18 years, RCT, Meta-analyis, Review.

The systematic literature review identified 46 potential articles on PubMed and 66 on Scopus. After having excluded the articles in common, the total number of articles was 88, and 60 papers were excluded after screening the titles or the abstracts due to not fitting with our topic. Thus, the actual review includes 28 articles, selecting the most recent and relevant for the argument.

## 2. Atopic Dermatitis

AD or atopic eczema is the most common chronic and recurrent inflammatory skin disorder in children [13]. The AD prevalence has risen during the last few years, especially in industrialized countries, reaching up 20% prevalence in infant population around the world [14].

AD occurs at any age, though 45% of all cases begin within six months of life, with up to 80–90% developing their first symptoms by five years of age [15]. Infants with AD are predisposed to developing food allergy, allergic rhinitis, and asthma later in life, a process called the atopic march [16,17].

### 2.1. Pathophysiology

The pathogenesis of the disease is complex and multifactorial involving skin barrier failure, local and systemic immune dysregulation, gut, and skin dysbiosis and also genetic factors interacting with each other [18,19]. Skin barrier defects may be linked with mutations of filaggrin gene, a structural protein expressed in the stratum corneum of the skin [20]. In particular, homozygous mutations of filaggrin gene are related to an augmented risk of severe AD, earlier onset and longer persistence of the disease [21]. Moreover, other factors, such as a reduction in cutaneous ceramides, lead to trans-epidermal water loss and an enhanced penetration of irritants, allergens, and microbes into the skin [22]. Skin barrier disruption is responsible for chronic inflammation with epidermal hyperplasia and cellular infiltrates (dendritic cells, eosinophils, and T-cells) [23]. In particular, an over-expression of T cells leads to the release of chemokines and pro-inflammatory cytokines that promotes IgE production as well as local and systemic inflammation (Figure 1) [24].

Lastly, microbiota abnormalities are present in patients with AD, showing a low degree of diversification in terms of bacterial colonization [25]. Furthermore, it has been demonstrated that there is an overgrowth of skin bacteria, such as *Staphylococcus aureus* [26], which colonizes about 90% of these patients, and *Corynebacterium* species in conjunction with a reduction of *Streptococcus*, *Propionibacterium*, *Acinetobacter*, *Corynebacterium*, and *Propionibacterium* [27]. Moreover patients with AD have gut microbial alteration showing increased *Escherichia coli*, *Clostridium difficile,* and *Staphylococcus aureus* and a decrease of beneficial microbes, such as *Lactobacillus* and *Bifidobacterium* (Figure 2) [28].

### 2.2. Clinical Aspects and Diagnosis

AD is a chronic condition with variable clinical phenotypes in relation to age, disease severity, and ethnicity. Moreover, even milder forms have a detrimental effect on life quality for both patients and their relatives [29]. In roughly 60% of cases, AD develops in the first year of life (early onset) and the disease can be either relapsing-remitting or persistent. The earliest clinical signs are skin dryness (xerosis) and roughness, so, typically pruritus. Acute lesions are characterized by diffuse erythematous patches and exuding papule-vesicles. Subacute injuries appear red and dry. Chronic lesions are sparsely demarcated and have scaly patches and plaques with excoriation and lichenification. Lesions can occur in any part of the body, but typically show age-related morphology and distribution. In infants, AD is generally acute, with lesions mainly on the face and the extensor surfaces of the limbs and the trunk Among adolescents and adults the lesions are often lichenified and excoriated plaques at flexures, wrists, ankles, and eyelids; in the head and neck type, the upper trunk, shoulders, and scalp are involved. Adults might only have chronic hand eczema or present with prurigo-like lesions [30]. To measure disease severity several methods have been established, recognizing symptoms, objective signs, quality of life, and long-term control as core domains, e.g., the Eczema Area Severity Index (EASI), Scoring Atopic Dermatitis (SCORAD) index, Numeric Rating Scale (NRS), and Dermatology Life Quality Index (DLQI) [31,32,33]. For the lack of specific diagnostic tests or laboratory biomarkers, characteristic clinical features, disease evolution, and personal and family history are pivotal for AD diagnosis [34].

### 2.3. Treatment

AD management aims to reduce symptoms and improve long-term disease control. General measures are important to prevent AD exacerbations, including wearing loose-fitting cotton clothing, the avoidance of overly heated environments, taking showers of a shorter length and avoiding hot water, washing without the use of soap. In addition, the regular use of emollients, which provide an occlusive layer and thereby limit evaporation, is useful in reducing the need for topical corticosteroids. The main drugs of anti-inflammatory therapy are topical corticosteroids that should be used until the resolution of lesions, calcineurin inhibitors (Tacrolimus, Everolimus, Pimecrolimus) allowed in children older than two years with moderate–grave AD, and systemic immunosuppressors, such as Cyclosporine, Azathioprin, and Methotrexate, used in patients who do not respond to standard therapies. Dupilumab, a monoclonal antibody directed against IL-4α and IL-13 receptors, has been approved by the FDA for the treatment of moderate to severe AD forms among adults and children older than 12, demonstrating a significantly reduction of clinical manifestations of moderate to severe AD forms [35]. Anti-infective measures such as antibiotics are often necessary to treat exacerbations, consequences of a microbial colonization of the skin. To prevent the role of bacteria in the pathogenesis or in its exacerbation, recent studies are testing the efficacy of probiotics in treating patients with AD [36].

## 3. Probiotics

The United Nations Food and Agriculture Organization (FAO) and World Health Organization (WHO) have defined probiotics as “live microorganisms, which when administered in adequate amounts confer a health benefit on the host” [37]. Probiotics include different strains and species of microorganisms with a broad and diverse range of clinical and immunologic capacities. *Lactobacillus* and *Bifidobacterium* are the most frequently strains utilized as probiotics, but other species, including the yeast *Saccharomyces Boulardii* and some *E. coli* and *Bacillus* species, are also used [38]. The capacity to survive in the intestine, to adhere to the gastrointestinal mucosa, and to compete with pathogens are among the criteria required to be defined as probiotics [39]. Diet, environmental conditions, exposure to probiotics along with many other host factors influence microbiota composition that is both unique and dynamic [40]. The intestinal microbiome is considered increasingly relevant for maintaining health status and for the intervention in numerous diseases such as inflammatory bowel diseases (IBD), irritable bowel syndrome (IBS), acute antibiotic-associated and *Clostridium difficile*-associated diarrhea, necrotizing enterocolitis (NEC), and *Helicobacter pylori* infection [41]. The biological mechanisms by which probiotics offer beneficial effects in the host include enhancement of the barrier function, suppression of pathogens, and modulation of the immune system [42].

### 3.1. Enhancement of Barrier Function

Probiotics improve intestinal barrier function through various mechanisms. They include the promotion of mucin secretion by goblet cells trough upregulation of mucin-type glycoprotein (MUC)1, MUC2, MUC3, and the consequent limitation of bacterial movement through the mucus film, increasing the secretion and expression of antimicrobial peptides (AMPs), such as α-defensin and β-defensin, which prevents bacterial proliferation and enhancement of tight junction (TJ) stability via the upregulation of transmembrane TJ protein (claudin-1, occludin) and intercellular TJ protein (zonula occludens (ZO)), resulting in a decrease epithelial permeability to pathogens and their products [43].

### 3.2. Suppression of Pathogens

Probiotics compete with pathogens or commensals for binding sites on mucins or epithelial cells and preventing overgrowth of potentially pathogenic microorganisms. In addition, probiotics provide an antimicrobial factor such as antimicrobial peptides, short-chain fatty acids (SCFA), and bacteriocins that are involved in suppressing or killing pathogenic organisms. Furthermore, SCFA such as butyrate, for instance, assists in modulating the expression of occludin and ZO both of which are involved in the enhancement of epithelial barrier integrity [44]. Probiotics are also able to increase IgA production in the host’s gastrointestinal (GI) tract. The secretory IgA protect the intestinal epithelium against colonization and/or invasion occurring through the link of antigens of pathogens or commensal, inducing the retro-transport of antigens to dendritic cells (DC) and the down-regulation pro-inflammatory responses [45].

## 4. Probiotics Function on Preventing AD

The immune system includes innate and adaptive immunity, which work together. The innate immune reacts immediately to infectious agents representing the first line of defense against pathogens. The protagonists in the innate immune response are surface barriers, specialized phagocytes (neutrophils, monocytes, and macrophages), soluble factors, DC, as well as natural killer (NK) cells that rapidly react to the presence of virus-infected cells by killing the infected target cell [46]. Pathogen-associated molecular patterns (PAMPs) are recognized by the cellular receptors on the surface of immune cells called pattern recognition receptors (PRRs). They include Toll-like receptors (TLRs) expressed mainly by macrophages and DCs, Nod-like receptors (NLRs), G protein-coupled receptors (GPGRs), and the aryl hydrocarbon receptor (AHR) [47]. The involvement of PRRs results in cellular activation. Activation of DC causes maturation and cytokines production that also influence the adaptive immune system, particularly the polarization of T-cell responses [48]. The adaptive immune system because of their ability to recognize and remember an impressive number of antigens, can provide a more effective protection against pathogens. Key players in adaptive immunity are lymphocytes B and T. B cells contribute to the immune response by secreting antibodies (humoral immunity), whereas T cells act in cell-mediated immunity. T cells include T helper cells (CD4+) and T cytotoxic cells (CD8+) [46]. Adaptive immune response is induced by the activation of antigen presenting cells (APC). DC are the major APC and they play a central role in regulating immune response. Differentiation of immature DC into mature or tolerogenic DC occur in presence of anti-inflammatory stimuli such as transforming growth factor (TGF)-β, retinoic acid and IL-10. Matured DC promote T cells differentiation towards Th1 or Th2 phenotypes with polarized cytokine secretion. Matured DC interact with naïve T cells (CD4+) and depending on the resulting cytokine produced by the CD4+ cells they differentiate to different Th subsets, promoting either an inflammatory response (Th1, Th2, Th17) or a regulatory one (Treg). Th1 differentiation occur in presence of IL-1, Il-6, IL-12, IFN-γ, and TNF-α. These cytokines activate macrophages, induce killed mechanisms, including cytotoxic cells, upregulate tumor immunity and intracellular pathogens, and are involved in autoimmunity. Th17 differentiation happens in the presence of IL-17A, Il-17F, IL-21, IL-22, and IFN-γ. Th2 differentiation occurs in the presence of IL-4, IL-5, IL-13, IL-9, and IL-6 and is involved in allergic reactions, allowing the production of allergen-specific IgE by B cells helping to induce eosinophil production [49,50]. As a result of IgE attachment to the Fc receptor on the surface of mast cells and basophil cells, the degranulation of the latter occurs, thereby contributing to the inflammation and the symptoms of allergies. Moreover, tolerogenic DC stimulate Treg cells in the presence of IL-2 and TGF-β. Treg cells represent a specialized T cell subpopulation determinant for immune homeostasis maintenance regulating Th1 and Th2 balance. They down-modulate IgE synthesis, reduce allergic reactions, and are responsible for the state of unresponsiveness of the immune system to self- peptides [51]. Treg cells include ‘natural’ Treg cells (nTreg) and, ‘inducible’ Treg cells (iTreg), which are subdivided into: induced Treg cells, type 1 regulatory T cells (Tr1), TGF-β expressing Th3, IL-17 producing FoxP3 Treg cells, CD8+Treg cells, double negative CD4 CD8 TCRαβ+Treg cells, and TCRγδ Treg cells [52]. A subset of DCs, expressing the integrin chain CD103+DCs, is the major population of DCs carrying antigen from the intestine to the mesenteric lymph nodes where they induce the differentiation of naïve T cells into Tregs. CD103+DCs are considered major drivers of tolerance in the intestine and are crucial for inducing Treg cells. They are particularly efficient to metabolize vitamin A to retinoic acid. Vitamin A conversion is essential for CD103+DCs to maintain their tolerance-inducing phenotype, including through the activation, expansion, and gut homing of Tregs [53]. An important inducer of allergen tolerance is forkhead box P3 (FOXP3), a transcription factor essential for Treg function. It is mainly expressed in a subset of CD4+ T-cells that play a suppressive role in the immune system. Many factors such as cytokines and non-cytokine factors regulate the generation of FOXP3^+^ T-cells. For example, retinoic acid, produced by the dendritic cells and epithelial cells in the intestine, works together with TGF-β1 and promotes the generation of small intestine-homing FOXP3^+^ T-cells by upregulating the expression of FOXP3 and gut homing receptors [54], a transcription factor expressed by FOXP3+Treg cells together with CD25 as well as IL-10 and TGF-β production, which are cytokines that in general suppress immune responses [55]. The disruption of Th1/Th2 balance, which results in a prevalence of Th2 cell subset and their secretory cytokines, is the cause of the development of allergic disease [56]. Probiotic bacterial cells, as well as most antigens and commensal bacteria, reach the intestinal lumen by M cells and DC. Probiotics exert their immunomodulatory effects on allergic diseases, balancing the Th1/Th2 immune response, stimulating Th1 and decreasing Th2 response through different cytokines secretion [57]. They act by various pathways: (a) promoting the differentiation of immature DCs into mature or tolerogenic DCs in presence of anti-inflammatory cytokines, such as IL-10, TGF-β; (b) inducing the differentiation and proliferation of Tregs cells via the induction of CD4+ Foxp3+ cells and CD103+DCs in the presence of IL-2 and TGF-β (Figure 3) [58].

Moreover, probiotics act on a reduction in allergen specific IgE and they also aid homeostasis by maintaining intestinal epithelial integrity, increasing antimicrobial production, and competitively inhibiting the survival of pathogens and increasing the production of secretory IgA [59]. Furthermore, intestinal microorganisms exert their immune-modulating action through several metabolites resulting from undigested carbohydrates complex fermentation, such as SCFA, aryl hydrocarbon receptor (AHR) ligands, and polyamines. These are crucial to preserve immune homeostasis in the gut regulating protective and inflammatory responses [60]. SCFA, including butyrate, propionate, acetate, and pentanoate, have been shown to have inhibitory effects on histone deacetylases (HDAC), modulating the expression of different genes involved in several biological processes, such as cell proliferation and differentiation and this may promote the development of peripherally induced Treg cells. The expansion and differentiation of Treg cells are induced by SCFA, especially by butyrate [61]. Butyrate promotes anti-inflammatory pathways by inducing CD4+Tcell differentiation into Treg cells mediated by HDAC through the G protein-coupled receptors 109A (GPR109A) expresses on the surface of these cells. This receptor act on CD103+ cells to promote Treg proliferation and expansion in mesenteric lymph nodes. [62]. In addition, SCFAs induce gut DC to express retinal aldehyde dehydrogenase (RALD), which produces retinoic acid from vitamin A [63] and would promote Treg cell differentiation [64]. Finally, butyrate seems to be able to suppress degranulation triggered by the binding of IgE to the high-affinity IgE receptor of mast cells, thus leading to a reduction in the release of inflammatory mediators and histamine, reducing the development of allergic reaction [65]. In addition, SCFA, enhancing acetyl-CoA and oxidative phosphorylation and glycosyl and fatty acid synthesis, increase antibody production [66]. It has been proven that the embryo receives the maternal microbiota that translocates through the vagina, maternal gut, and placenta. The translocation from the maternal gut and placenta occurs in the bloodstream after dendritic cell-facilitated translocation across the gut epithelium while meconium microbes result from the amniotic fluid that is swallowed [1]. Later, the type of feeding (breastfeeding or formula feeding) affects the newborn microbiota. An entero-mammary pathway allows the transfer of microbes from the maternal gut to the mammary gland. Bacterial translocation occurs from the gastrointestinal tract into lamina propria, and then to the mesenteric lymph nodes and bloodstream [67]. Probiotics administered to mothers during pregnancy and lactation first reach the fetus and then the newborn through the same pathways described above. The bacterial strains exert their action after reaching the child’s gastrointestinal tract.

## 5. Human Studies

Several studies have focused on AD prevention using probiotics in children with conflicting data regarding the outcome (Table 1) [68,69,70,71,72,73,74,75,76,77,78,79,80,81,82,83,84,85,86,87,88,89,90,91,92,93,94]. We have performed a review of the most relevant and latest articles on this topic.

### 5.1. Monostrain

In 2011, Boyle et al. conducted a randomized controlled trial assessing the impact of *Lactobacillus rhamnosus* on atopic dermatitis prevention in infants when administrated in pregnant women [68]. The probiotic supplementation was related only to 250 pregnant women from 36 weeks of gestation until delivery, and these women were selected as a result of the high risk of allergic disease for their children. Unlike the majority of other studies, the authors found no statistically significant difference between the probiotic supplemented group and the placebo group on the cumulative incidence of eczema (34% probiotic, 39% placebo; RR 0.88; 95% CI 0.63, 1.22) or IgE-associated eczema (18% probiotic, 19% placebo; RR 0.94; 95% CI 0.53, 1.68) during the first year in offspring. In a double-blind randomized placebo-controlled trial published in 2012, Wickens et al. investigated the cumulative prevalence of eczema in children at age four years and two years after stopping the probiotic supplementation with *Lactobacillus rhamnosus* HN001 (6 × 10^9^ cfu/day) or *Bifidobacterium animalis* subsp *lactis* HN019 (9 × 10^9^ cfu/day) [69]. They supplemented mothers of infants at high risk from 35 weeks of gestation until six months if breastfeeding and their newborn until two years. They followed-up this cohort of infants with SCORAD for eczema until four years, showing that the cumulative prevalence of eczema was significantly reduced in the group receiving HN001 (HR 0.57, 95% CI 0.39–0.83). Enomoto et al., in an open trial published in 2014, highlighted that the supplementation, in the prenatal and postnatal period, with two species of *Bifodobacterium* (*Bifidobacterium breve* M-16V and *Bifidobacterium longum* BB536) reduced the risk of developing eczema and AD in infants. After 18 months of follow-up, the authors observed a lower incidence of atopic eczema in the probiotic group (OR: 0.304 [95% CI: 0.105–0.892] [70]. In 2017, Cabana et al. conducted a study evaluating the effect of probiotic administration during the first six months of life on childhood eczema. The randomized, double-blind controlled trial enrolled a total of 184 infants (92 in the probiotic group and 92 in the placebo group). These infants were supplemented with *Lactobacillus rhamnosus* GG (LGG) during their first six months of life. The aim of the study was to determinate the cumulative incidence of eczema (primary endpoint) in such high-risk infants. The authors observed the children for a median follow-up period of 4.6 years and assessed at two years of age a cumulative incidence of eczema of 30.9% (95% CI, 21.4–40.4%) in the control group and 28.7% (95% CI, 19.4–38.0%) in the LGG group, with an HR of 0.95 (95% CI, 0.59–1.53) (log-rank *p* = 83). They concluded that, for high-risk infants, an early monostrain probiotic supplementation does not prevent the AD development when they were evaluated at two years of age [71]. In a two-center RDBCT in 2018, Wickens et al. performed the first study which evaluated early probiotic intervention with positive results for at least the first decade of life in children. The authors administrated a single strain of probiotic (*Lactobacillus rhamnosus* HN001 (HN001) (6 × 109 colony-forming units [cfu]) or *Bifidobacterium lactis* HN019 (HN019) (9 × 109 cfu) to pregnant women from 35 weeks of gestation to six months post-partum if breast feeding and from birth to age two years in infants. In this study, Wickens et al. followed up their patients for 11 years and they demonstrated that *Lactobacillus rhamnosus* HN001 significantly protected against the development of eczema (relative risk [RR] = 0.46, 95% CI 0.25–0.86, *p* = 0.015), while there was no protective effect of *Bifidobacterium lactis* HN019 [72].

### 5.2. Multistrain

A randomized, double-blind trial, conducted by Dotterud et al. and published in 2010, showed a reduction of incidence of atopic dermatitis at two years of age in children of non-selected mothers receiving probiotic milk from 36 weeks of gestation to three months postnatally during breastfeeding [73]. This study aimed to investigate the role of probiotic milk containing *Lactobacillus rhamnosus* GG, *L. acidophilus* La-5, and *Bifidobacterium animalis* subsp. *lactis* BB-12 for the prevention of allergies. In particular, the odds ratio (OR) for the incidence of AD in the probiotic group was 0.51, compared with the placebo group (95% CI 0.30–0.87, *p* = 0.013). In addition, in stool samples of children, only the presence of the *Lactobacillus rhamnosus* GG strain was detected, despite the administration of three strains during pregnancy and lactation, suggesting a different transmission capacity of different strains from mother to child. In a cohort study published in 2013, Randi et al. examined the association between probiotic milk consumption (milk containing *Lactobacillus acidophilus LA-5, Bifidobacterium lactis* BB-12, *Lactobacillus rhamnosus*) during the pregnancy and infancy period and the relative incidence of allergies diseases in children [74]. They calculated relative risks, reported by questionnaire, for atopic eczema, rhino conjunctivitis and asthma in a large cohort of children. The authors concluded that probiotic milk consumption was related to a decreased incidence of atopic eczema and rhino conjunctivitis, without any association with a maternal history of allergic disease. In particular, the relative risk (RR) of atopic eczema at six months was 0.94 (95% CI, 0.89–0.99) if probiotic milk was consumed during pregnancy. In a randomized controlled trial published in 2014, Allen et al. evaluated the preventive effect on eczema in children of a multistrain, high-dose probiotic supplementation in pregnant women and their infants [75]. In this study, women from 36 weeks gestation and their infants up to age six months received daily a mixture of probiotic (*Lactobacillus salivarius* CUL61, *Lactobacillus paracasei* CUL08, *Bifidobacterium animalis* subspecies lactis CUL34 and *Bifidobacterium bifidum* CUL20). Infants were followed-up for eczema until they reached two years of age. The authors found no difference of incidence for eczema between the probiotic and the placebo arms. However, the cumulative frequency of skin prick sensitization to common food allergens was reduced in the probiotic group. In a RDBCT of Simpson et al. published in 2015, the authors demonstrated that maternal probiotic supplementation alone may be sufficient for reduction in the cumulative incidence of AD in children in the long term. In this study, Simpson et al. randomized 450 pregnant women to receive placebo or probiotic milk from 36 weeks of gestation until three months post-partum. This was a multistrain probiotic study, evaluating the effect of a probiotic milk containing *Lactobacillus rhamnosos* GG, *Lactobacillus acidophilus* La-5 and *Bifidobacterium animalis* subsp. *lactis* BB-12. After a follow-up period of six years, the children were evaluated for the presence of AD. A lower cumulative incidence of AD in the probiotic group (OR 0.64, 95% CI 0.39–1.07, *p* = 0.086; NNT = 10) was assessed compared to the placebo group [76]. In 2018, Schmidt at al. published a double-blind, placebo-controlled trial evaluating the effect of a mixture of two probiotic strains (*Lactobacillus rhamnosus* and *Bifidobacterium animalis* subsp *lactis)* applied in late infancy and early childhood on the development of allergic diseases. A total of 290 infants (144 in the probiotic group and 146 in the placebo group) were randomized to receive a daily mixture of probiotics or placebo for a period of six months. The participants were evaluated monthly with web-based questionnaires on allergic symptoms, medical diagnosis of allergic disease and serum IgE levels. At follow-up, the authors observed a significantly lower incidence of eczema in the probiotic group (4.2%) compared to the placebo group (11.5%), assessing a protective role of probiotics on the development of AD with a relative risk of 0.37 (95% CI 0.14–0.98; *p* = 0.036) [77].

### 5.3. Review and Meta-Analysis

In 2011, Doege et al. published a meta-analysis evaluating the role of probiotics supplementation during pregnancy on the development of eczema in children. The authors considered a total of seven randomized, double-blind, placebo-controlled trials, published between 2001 and 2009. The completed meta-analysis highlights a significant risk reduction for atopic eczema in children aged 2–7 years after their mothers had been supplemented with the probiotics during pregnancy (reduction 5·7%; *p* = 0·022). In particular, this effect was significant for lactobacilli (reduction 10·6%; *p* = 0·045). On the other hand, a mixture of different bacterial strains as probiotics does not have the same effect on eczema prevention in children (difference 3·06%, *p* = 0·204) [78]. In 2012, Pelucchi et al. published a meta-analysis supporting the use of probiotics during pregnancy or early life in children for the prevention of AD and Ig-E associated AD in infants [79]. In particular, the authors analyzed randomized controlled trials updated to October 2011 and reported a reduction of about 20% in the incidence of AD and IgE-associated AD in infants and young children following probiotic use. In the 14 trials examined with a systematic literature search, probiotics were given according to several intervention regimens, to pregnant women in some studies and to infants at weaning in other studies. The primary outcome of the study was the demonstration of a decreased incidence of both AD, with a RR of 0.79 (95% CI 0.71–0.88), and IgE-associated AD, with a RR of 0.80 (95% CI 0.66–0.96). In 2014 Mansfield et al. conducted a review to analyze the impact of prenatal and postnatal probiotic supplementation on eczema prevention in infancy and childhood. The authors concluded that the use of probiotic supplements during pregnancy and/or during the first six months of life reduces significatively the incidence of eczema in infants and children. In particular, it has been shown to decrease the incidence of eczema in infants by 26% (18–33%). Meta-analysis of in utero administration of probiotics proved statistically significance with a RR of 0.77 (95% CI 0.64, 0.93), while studies with postnatal exposure were less conclusive, limiting the statistical power of the comparison. In protecting eczema, the most efficacious strains of probiotics were Bifidobacterium, but strains of *Lactobacillus* also showed a protective effect [80]. A meta-analysis published in 2015 by Cao et al. evaluated the long-term (no less than five years) effect on preventing AD if probiotics were administrated in early life, estimated as 14% compared with the placebo (*p* = 0.005) [81]. A total of six randomized, double blind and placebo-controlled trials (including a total of 1955 eligible patients) were included in the study. The meta-analysis supported the evidence of a strong association between the consumption of probiotics in early life with the prevention of AD in the long term, with an RR of 0.86 (95% CI 0.77–0.96, *p* = 0.005). Sub-analysis highlighted the important role of both prenatal and postnatal administration (*p* = 0.002), rather than only postnatal administration (*p* = 0.89), in decreasing the cumulative incidence of atopic eczema. A systematic review and meta-analysis by Zuccotti et al. published in 2015 examined randomized-controlled trials evaluating the administration of probiotics to pregnant women and/or to infants in the first three months of life, and the effect of their administration in preventing allergic disease in high-risk children [82]. The authors analyzed a total of 17 studies, including data from 4755 children (2381 in the probiotic group and 2374 in the control group). This study showed that infants treated with probiotics had a reduced incidence of eczema in the first 24 months of life compared to controls (28.22% versus 35.67%; RR 0.78; 95% CI 0.69–0.89; *p* = 0.0003), with a partial loss of efficacy after two years of life. Moreover, a sub-meta-analysis showed that those supplemented with probiotic mixtures rather than with either *Lactobacilli* or *Bifidobacteria* alone, had better results in the prevention of eczema (RR 0.54; 95% CI: 0.43–0.68; *p* < 0.00001). In 2015 the World Allergy Organization (WAO) joined a guideline panel to develop evidence-based recommendations to prevent allergies with the use of probiotics. The authors reviewed 23 randomized controlled trials, including the use of probiotics only in infants, in pregnant women and infants and finally among pregnant women, breastfeeding mothers and infants [83]. According to the studies analyzed, the WAO guideline concluded that probiotics assumed by pregnant woman provide clear advantage, especially for the prevention of eczema in high-risk infants, although the evidence was low; equally, probiotics reduced the rate of eczema when compared to placebo (RR 0.61, 95% CI from 0.50 to 0.64), if administrated in breastfeeding mothers and infants (RR 0.81, 95% CI from 0.71 to 0.94). In their systematic review published in 2015, Cuello-Garcia et al. evaluated randomized-controlled trials that analyzed probiotic supplementation in pregnant women, breastfeeding mothers, infants, and children [84]. This systematic review, including a total of 29 randomized trials comparing at least one probiotic with placebo, assessed a reduced risk of eczema in infants with probiotic use during pregnancy (RR 0.72; 95% CI 0.61–0.85), during breastfeeding (RR 0.61; 95% CI: 0.50–0.74) and in infancy (RR 0.81; CI: 0.70–0.94). Hulshof et al. published a review which included all the clinical trials from the 2008 to 2017 with the goal of defining the role of microbial modulation in AD management. However, despite the vast numbers of studies evaluated, the wide heterogeneity of these made it difficult to reach the primary aim [85]. Another review conducted in 2018 by Sharma et al., who wanted to evaluate the immunomodulatory potential of probiotics in allergic disease. They concluded by reinforcing their beneficial effects in AD prevention, however without writing definitive conclusions [86].

In 2018, Lin Li et al., in their systematic review and meta-analysis, in the 28 articles included, found that probiotic supplementation during both prenatal and postnatal period reduced AD incidence in infants and children. So, the benefit in AD prevention is linked to the use of probiotic treatment during gestation and the first six months of the infant’s life [87].

They observed that the use of mixtures of probiotics including strains of *Lactobacillus*, *Bifidobacterium* and *Propionibacterium* all appeared to reduce the incidence of AD during both the prenatal and postnatal period, from birth to six months of age (OR 0.67; 95% CI: 0.59–1.01), or only post-natal use (OR 0.66; 95% CI: 0.37–1.15). Moreover, receiving probiotics no more than six months after birth was shown to reduce the incidence of AD (OR 0.61; 95% CI 0.48–0.76), but administering probiotics for >12 months had no effect in preventing AD, compared to controls. A more recent systematic review was conducted in 2019 by Petersen et al., according to PRISMA guidelines, on the function of gut microbiota in AD. After analyzing 44 studies, they concluded about the controversial role of gut microbiome on the onset and severity of AD [88]. In 2019, Yang et al. conducted a meta-analysis to evaluate probiotics supplementation’s rule in the prevention of eczema in children if administrated during pregnancy and infancy. This investigation concluded that there is still controversy regarding the use of probiotics in pregnant woman or children with high-risk eczema, in particular concerning which probiotics play the major role in preventing eczema, the dose of effective probiotic, and the best time of the administration [89].

Amalia et al., in 2019, elaborated a systematic review and meta-analysis on the use of probiotic supplementation in pregnant and breastfeeding mother as well as in infant AD prevention in children. They showed, after considering 21 articles, that a mixture of probiotic supplementation given to the mother in pregnancy and continuing while breastfeeding, and also to high-risk infants, was the most efficacious strategy to reduce AD risk in children [90].

In 2020, Jiang et al., in their systematic review and meta-analysis of RCT, after analyzing 14 prevention studies, they concluded that probiotics administered only to infants did not prevent the AD. They observed an effect when administered to both pregnant mothers and their infants or only to pregnant mothers [91].

The systematic review conducted by Debra de Silva et al. in 2020, using the GRADE approach, on the prevention of FA in infants and children, did not detect any effect of the role of probiotics in FA prevention [92]. In 2020, the meta-analysis conducted by Tan-Lim et al., considering 21 randomized controlled trials, stated that the best probiotic preparations able to knock down the risk of AD are Mix 8 (*Lactobacillus paracasei* ST11, *Bifidobacterium longum* BL999) (RR = 0.46,53 95% CI 0.25–0.85) and Mix3 (*Lactobacillus rhamnosus* GG, *Bifidobacterium animalis* ssp *lactis* BB-12) (RR = 0.50, 95% CI 0.27–0.94). In addition, *Lactobacillus rhamnosus* HN001 administered during pregnancy and early infancy seems to reduce the risk of AD (RR = 0.54,64 95% CI 0.26–1.11) [93]. A recent meta-analysis published in 2021 by Sun et al. investigated the impact of a mixed strain of *Lactobacillus* and *Bifidobacterium* on eczema in infants younger than three years old [94]. The authors selected nine randomized double-blind placebo-controlled trials, with a total of 2093 infants, where pregnant women and/or breastfeeding mothers or infants younger than three years old assumed oral *Lactobacillus* and *Bifidobacterium* mixed strains for the prevention of eczema. They effectively demonstrated that the *Lactobacillus* and *Bifidobacterium* mixed strain could prevent eczema, compared to the placebo. Subgroup analysis disclosed that the mixture of two probiotic strains had a preventive impact on both infants with positive (RR = 0.53; *p* < 0.001) and negative (RR = 0.69; *p*: 0.02) family history. The effect of probiotics mixture in early pregnancy was more significant (RR = 0.59; *p* < 0.001), compared with the intervention of infants alone. Furthermore, the authors highlighted that a daily dose of probiotics ≤ 1 × 109 and > 1 × 109 CFU is effective in reducing AD incidence (*p* < 0.01).

## 6. Conclusions

Several studies have been conducted for AD prevention in children with probiotics, administering them both to their mother in pregnancy and to the child in the first months of life. The results are encouraging, even if the results cannot be compared easily given the diversity in the type, dose, and timing of probiotics administration as well as the period of follow-up post treatment. Although the indications concerning the administration of probiotics in order to prevent of AD are currently approved by the WAO, further careful follow-up studies that are more high-quality and long-term are required in order to support the current evidence before the routine use of such probiotics is recommended.

## Figures and Tables

**Figure 1 ijms-23-05409-f001:**
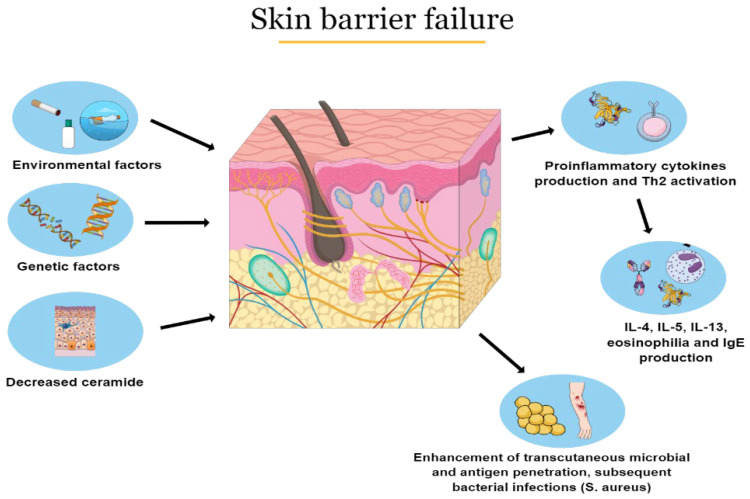
Pathophysiology of Atopic Dermatitis.

**Figure 2 ijms-23-05409-f002:**
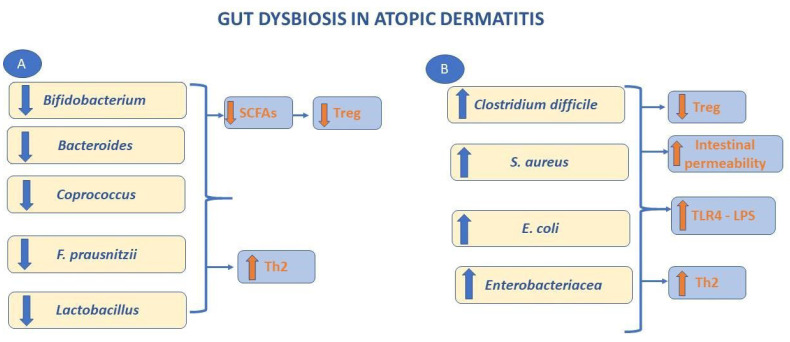
Gut dysbiosis in atopic dermatitis. The image shows gut microbial alteration in patients with AD. It seems that they have reduction of the beneficial microbes such as *Lactobacillus*, *Bifidobacaterium* (**A**) and an increase of the proportions of *E. coli*, *C. difficile* and *S. aureus* (**B**). Metabolites of *Bifidobacterium*, *Lactobacillus*, etc. are able to suppress the expression of Th2 associated cytokines. The reduction of bacteria producing SCFAs leads to an inadequate production of Treg cells. *E. coli* can promote an intestinal inflammatory response thanks to LPS. *C. difficile*, *S. aureus* could be associated with increase of intestinal permeability and eosinophilic inflammation. AD: atopic dermatitis; SCFAs: short chain fatty acids; Treg: regulatory T; TLR4: tool like receptor 4; LPS: lipopolysaccharide.

**Figure 3 ijms-23-05409-f003:**
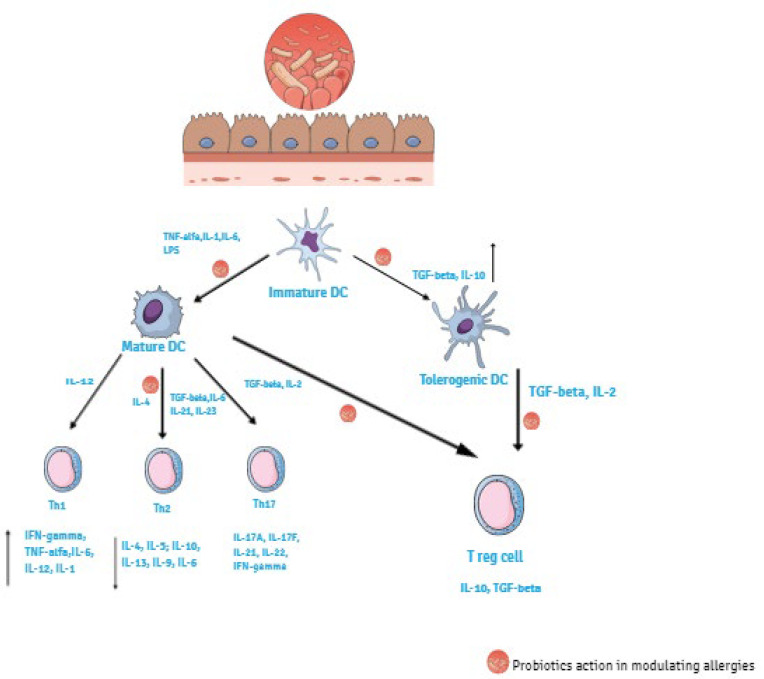
Immunomodulatory effects of probiotics. Probiotics regulate the differentiation of immature DCs into mature or tolerogenic DCs in the presence of pro-inflammatory stimuli (TNF-α, IL-1, IL-6, LPS) or anti-inflammatory stimuli (TGF-β, IL 10) respectively. Inflammatory Th1 cells are produced in presence of Il-1, IL-6, IL-12, IFN-γ, TNF-α, instead anti-inflammatory Th2 cells in presence of IL-4, IL-5, IL-9, IL-10, IL-13, IL-33. Probiotics exert their immunomodulatory effects on allergic diseases balancing Th1/Th2 immune response stimulating Th1 and decreasing Th2 response through different cytokines’ secretion and inducing beneficial Treg cells to promote immune tolerance.

**Table 1 ijms-23-05409-t001:** Main clinical studies performed with oral probiotics for atopic dermatitis prevention in children.

Author, Year, Nationality	Study Design	Sample Size at Baseline	Sample Size at Follow-Up	Probiotics	Period of Administration	Follow-Up	Results
Dotterud et al. 2010, Norway [73]	RCT	AG: 138 mothers CG: 140 mothers	AG: 42 children CG: 58 children	Probiotic milk containing *Lactobacillus* *rhamnosus* GG-5, *Lactobacillus acidophilus* La-5 and *Bifidobacterium animalis* subsp. *Lactis* Bb-12	From 36 weeks of gestation to 3 months postnatally during breastfeeding	2 years	Reduction of incidence of AD at 2 years of age in children of mothers receiving probiotic milk (OR 0.51, 95% CI 0.30–0.87, *p* = 0.013)
Boyle et al. 2011, Australia [68]	RCT	AG: 125 mothers CG: 125 mothers	AG: 108 infants CG: 102 infants	*Lactobacillus rhamnosus* GG	From 36 weeks of gestation to delivery	12 months	No statistically significant difference between the active group and the placebo group on the cumulative incidence of AD (34% probiotic, 39% placebo; RR 0.88; 95% CI 0.63, 1.22) or IgE-associated AD (18% probiotic, 19% placebo; RR 0.94; 95% CI 0.53, 1.68)
Wickens et al. 2012, New Zealand [69]	DBRCT	AG: 157 infants (HN001 group) AG: 158 infants (HN019 group) CG: 150 infants	AG: 136 infants (HN001 group) AG: 146 infants (HN019 group) CG: 143 infants	*Lactobacillus rhamnosus* HN001 *Bifidobacterium animalis* subsp. *Lactis* HN019	From 35 weeks of gestation to 6 months of age after birth in mothers if breastfeeding. From birth to 2 years in all infants	4 years	The cumulative prevalence of AD was significantly reduced in the group of infants receiving HN001 (HR 0.57, 95% CI 0.39–0.83)
Enomoto et al. 2014, Japan [70]	Open trial	AG: 130 pregnant woman and their infants CG: 36 mothers-infant pairs	AG: 94 infants CG: 31 infants	*Bifidobacterium breve* M-16V and *Bifidobacterium longum* BB536	From 1 month prior to delivery to pregnant woman to 6 months after birth to infants.	18 months	After 18 months of follow-up, a lower incidence of AD in the probiotic group (OR: 0.304 [95% CI: 0.105–0.892])
Randi et al. 2014, Norway [74]	Cohort study	NA	AG: 15,042 infants CG:25572 infants	Probiotic milk containing *Lactobacillus acidophilus* La-5, *Bifidobacterium* subsp *lactis* BB12, *Lactobacillus rhamnosus*	From gestation in pregnant woman to 6 months after birth in infants	6 months	Consumption of probiotic milk products was related to a reduced incidence of AD in children (RR 0.94 [95% CI, 0.89–0.99])
Allen et al. 2014, UK [75]	RCT	AG: 220 mothers CG: 234 mothers	AG: 187 children CG: 191 children	*Lactobacillus salivarius* CUL61, *Lactobacillus paracasei* CUL08, *Bifidobacterium animalis* subsp *lactis* CUL34 and *Bifidobacterium bifidum* CUL20	From 36 weeks of gestation to 6 months of age in children	2 years	The probiotic seemed to prevent atopic sensitization to common food allergen but not to prevent AD in infants (OR 1.07 [ 95% CI 0.72 to 1.6])
Simpson et al. 2015, Norway. [76]	DBRCT	AG: 211 pregnant women CG: 204 pregnant women	AG: 81 children CG: 82 children	Probiotic milk containing *Lactobacillus rhamnosus* GG, *Lactobacillus acidophilus* La-5 and *Bifidobacterium animalis* subsp. *Lactis* BB-12	From 36 week gestation until 3 months postpartum in mothers	6 years	Perinatal maternal probiotic supplementation is effective in reducing the cumulative incidence of AD in children (OR 0.64 [95% CI 0.39–1.07, *p* = 0.086])
Cabana et al., 2017, California [71]	DBRCT	AG: 92 infants CG: 92 infants	NA	*Lactobacillus rhamnosus* GG	First 6 months of life	6 years	Early monostrain probiotic supplementation does not prevent the development of AD at 2 years of age (HR of 0.95 (95% CI, 0.59–1.53)
Wickens et al. 2018, New Zealand [72]	DBRCT	AG: 157 infants (HN001 group) CG: 158 infants (HN019 group) CG: 159 infants	AG: 97 children (HN001 group) AG: 104 children (HN019 group) CG: 97 children	*Lactobacillus rhamnosus* HN001 (HN001) or *Bifidobacterium lactis* HN019 (HN019)	From 35 weeks of gestation to 6 months’ post-partum in breastfeeding mothers; from birth to 2 years of age in infants	11 years	Lactobacillus rhamnosus HN001 significantly protected against the development of AD for at least the first decade of life (RR 0.46, 95% CI 0.25–0.86, *p* = 0.015)
Schmidt at al. 2018, Denmark [77]	DBRCT	AG: 144 infants CG: 146 infants	AG: 119 infants CG: 122 infants	*Lactobacillus rhamnosus* and *Bifidobacterium animalis* subsp *lactis*	From a mean age of 10 months for 6 months	6 months	Protective role of probiotics on the development of AD (RR 0.37, 95% CI 0.14–0.98; *p* = 0.036).

RCT: randomized controlled trial; AG: active group; CG: control group; AD: atopic dermatitis; DBRCT: double blind randomized controlled trial; NA: not available.

## Data Availability

Not applicable.

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
