# Peer review of "Probiotics Function in Preventing Atopic Dermatitis in Children"

_ijms, 2022, doi:10.3390/ijms23105409_

Round 1

Reviewer 1 Report

The article may be an useful contribution to the journal; however, several changes should be taken into consideration, as follows:

-originality:  As there are many studies and reviews already in the medical literature in the field, the authors should emphasise the novelty of their work, what they bring new to the already existing studies.

-link to the journal aims and scope: the molecular aspects have been discussed by authors only tangentially; in order to match the aim of the journal, molecular aspects need significant further elaboration, as an in-depth discussion needs would be in the benefit of the reader

-methodology: Line 245-246: authors state ‘We have performed a review of the most relevant and latest articles on this topic’- exact criteria for assessing relevance as well as time interval should be explicitly stated, preferable as a Methodology paragraph; this is important, as to minimise the risk of bias in the selection of the studies.

-content:  a few studies have not been included, such as: Jiang W, Ni B, Liu Z, Liu X, Xie W, Wu IXY, Li X. The Role of Probiotics in the Prevention and Treatment of Atopic Dermatitis in Children: An Updated Systematic Review and Meta-Analysis of Randomized Controlled Trials. Paediatr Drugs. 2020 Oct;22(5):535-549. doi: 10.1007/s40272-020-00410-6. PMID: 32748341. (article does not belong to the reviewer)

Moreover, please ascertain if study of Tan-Lim et al is from 2020 as authors state in Line 425-426; pls reference 84: Tan-Lim CSC, Esteban-Ipac NAR, Recto MST, Castor MAR, Casis-Hao RJ, Nano ALM. Comparative effectiveness of probiotic strains on the prevention of pediatric atopic dermatitis: A systematic review and network meta-analysis. Pediatr Allergy Immunol. 2021 Aug;32(6):1255-1270. doi: 10.1111/pai.13514. Epub 2021 May 15. PMID: 33811784. (article does not belong to the reviewer)

Authors are advised to rephrase ‘AD may happens at any age, but the incidence peak is in infancy, with 45% of all cases beginning within six months of life: in detail 60% of them during the first 73 year, 80-90% within fifth year of life [16] and this often represents’ as to clarify the numers and the percentages; the sentence is rather prolix and tedious; please rephrase, in the benefit of the reader.

-overall comments: Grammar and punctuation must also be carefully checked within the entire article (for example, but not limited to: line 72- ‘AD may happens at any age’)

Reviewer 2 Report

It is an review article dealing with the question of probiotics in patients with atopic dermatitis.

... Second-line agents in the treatment of AD are topical calcineurin inhibitors (Tacrolimus, Everolimus  + pimecrolimus

Authors should mention the atopic march and the occurrence of food allergy in patients suffering from AD.

I suggest to cite

Čelakovská, J. Bukač, R. Vaňková, J. Krejsek & C. Andrýs (2021) The relation between the sensitization to molecular components of inhalant allergens and food reactions in patients suffering from atopic dermatitis, Food and Agricultural Immunology, 32:1, 33-53, DOI: 10.1080/09540105.2020.1865281

 Čelakovská, I. Krcmova, J. Bukac & J. Vaneckova (2017) Sensitivity and specificity of specific IgE, skin prick test and atopy patch test in examination of food allergy, Food and Agricultural Immunology, 28:2, 238-247, DOI: 10.1080/09540105.2016.1258548

Reviewer 3 Report

This review is rather preliminary and lacks authors' viewpoints on the effects of probiotics on AD.

1. Line 14; regarding the incidence of AD10% in children while 20% in adults, is the percentage opposite?

2. Fig. 1; this figure does not correctly show that probiotics up-regulate Th1 versus Th2. Please revise this figure.

3. This review lacks schemes showing 1) the pathogenesis of AD, 2) gut dysbiosis in AD, and 3) how probiotics alter the pathogenesis of AD showing their actions on DC, Th2, Treg, or mast cells. These schemes should be incorporated.

4. Section 2; the description of general findings of AD not related to probiotics is too long, especially the treatment of AD.

5. The characteristics of gut microbiota in AD patients should be shown; low diversity? which genera are increased or decreased compared to healthy controls.

6. Line 185; occluding, means occludin? ZO-1 should be defined.

7. Please show the route of delivery of probiotics from a pregnant woman to fetus, are the probiotics taken by a woman delivered to fetus via placental circulation without modification? Similarly, the probiotics taken by a mother delivered into breastmilk via circulation without modification? These should be explained in detail.

8. How SCFA upregulate Treg should be more detailed including the effects on FOXP3 expression.

9. RCTs showing therapeutic effects of probiotics in children with AD should be presented,  not only of prophylactic effects.

Round 2

Reviewer 1 Report

The manuscript has been significantly improved, all suggestions have been taken care of, in an adequate manner.

Author Response

Please, see the uploaded file

Reviewer 3 Report

Line 338, IL-17 should be corrected to IL-17A. Since IL-17 family includes many sorts and thus should be specified.

Line 392: not TNF-beta, but TGF-beta. This should be corrected.

Fig. 3: just using star sign for the probiotics’ effects is  not appropriate. Please use up or down arrows. since probiotics reduce Th2 responses, this should be discriminated against Th1 or Treg.The author’s responses shown are not responses to me, but to another reviewer.

Please present the authors’ responses to my previous 9 points of criticism.

Without seeing that, I cannot evaluate the manuscript.

Author Response

Please, see the uploaded file

Round 3

Reviewer 3 Report

  1. Fig3: Finally, which of mature DC or tolerant DC do probiotics preferentially potentiate? Alternatively, do probiotics potentiate both? And what kind of mechanism works in this process? The authors did not describe this highly important point.

Furthermore, the authors did not describe the mechanism how probiotics preferentially potentiate Th1 while down-regulate Th2.

Besides the figure does not correctly show the above points.

  1. Fig 1: The authors arranged the parts of AD pathogenesis linking with arrows. However, Individual parts do not effectively link to the next parts. The explanation how each part lead to the next part is lacking.
  2. The figures in this article do not appropriately explain the concepts of the text, and the format is poor. The authors should consult with the other scientists familiar with writing this kind of reviews.
  3. The authors again failed to show the responses to my comments in the previous review. And thus I cannot know how they revised the manuscript. The authors’ responses shown are those to a different reviewer. There still exists the same error.
